# Attachment Theory: A Barrier for Indigenous Children Involved with Child Protection

**DOI:** 10.3390/ijerph19148754

**Published:** 2022-07-19

**Authors:** Peter Choate, Christina Tortorelli

**Affiliations:** Social Work, Mount Royal University, Calgary, AB T3E 6K6, Canada; ctortorelli@mtroyal.ca

**Keywords:** attachment theory, Indigenous child welfare, best interests of the Indigenous child, Indigenous parenting, child welfare

## Abstract

Background: Attachment theory is an established theoretical understanding of the intimate relationships between parental figures and children. The theory frames the ways in which a child can be supported to develop within a secure base that prepares them for adulthood, including entering into and sustaining intimate relationships. The theory, built on the work of John Bowlby following World War II, has extensive literature supporting its application across multiple cultures and nations, although its roots are heavily tied to Eurocentric familial understandings. However, the theory has also been heavily criticized as not being appropriate for child intervention decision-making. Further, its application to Indigenous caregiving systems is also under question. Yet courts rely heavily on applying the theory to questions of sustaining Indigenous children in non-Indigenous care when return to biological parents is deemed impossible. Methods: This article draws upon the consistent arguments used in leading Canadian child welfare legal decisions and case examples to show how Attachment Theory is applied relative to Indigenous children and families. Results: Attachment Theory drawing upon Eurocentric framing, and as applied in Canadian child protection systems, as seen in precedent court decisions, is given priority over living in culture. This occurs even though the research reviewed has shown that the traditional dyadic version of the theory is not valid for Indigenous peoples. Conclusions: While all children will attach to a caregiver or caregiving system, such as kinship or community, leading legal decisions in Canada tend to rely on Eurocentric versions of the theory, which is contrary to the best interests of Indigenous children. Child protection needs to reconsider how attachment can be used from appropriate cultural lenses that involve the communal or extended caregiving systems common to many Canadian Indigenous communities. Child protection should also recognize that there is not a pan-Indigenous definition of attachment and child-rearing, so efforts to build working relationships with various Indigenous communities will be needed to accomplish culturally informed caregiving plans. In addition, continued advocacy in Canada is needed to have child protection decision-making conducted by the Indigenous communities, as opposed to Eurocentric provincial or territorial agencies.

## 1. Introduction

In this paper, we examine the place of Attachment Theory as a significant framework that is used to keep Indigenous children separated from their culture, community, and kinship systems in Canada. This occurs as a result of the over-surveillance of Indigenous children, higher rates of removal from family and culture, and legal decisions that act to privilege Attachment Theory over cultural integration. We also explore how this pattern affects children, sustains inter-generational trauma (IGT), and possible pathways to address these issues.

In accordance with the goals of reconciliation in Canada from the period of Colonialism [1], it is important for researchers to socially locate themselves relative to Indigenous peoples and the legacy of colonization.

Peter Choate is a professor of social work at Mount Royal University in Calgary, Canada. He is from a white settler family system. He grew up on the traditional lands of the Musqueum, Tsel’ Waututh, and Squamish peoples.

Christina Tortorelli is an assistant professor at Mount Royal University in Calgary, Canada. She is the great-granddaughter of white settler families. She has lived most of her life in Calgary, which exists in Treaty 7 territory on the traditional lands of the Blackfoot peoples.

Both authors today live, work, and play on the ancient and storied places within the hereditary lands of the Niitsitapi (Blackfoot), Îyârhe Nakoda, Tsuut’ina, and Métis Nations. It is a land steeped in ceremony and history that, until recently, was used and occupied exclusively by peoples indigenous to this place. We speak from the place of the colonizers of Indigenous peoples in Canada.

## 2. Attachment Theory

Attachment Theory finds its roots in the works of John Bowlby. His early work was based upon his examination of children whom he characterized as having lost the bond with their primary caregiver [2]. He perceived that the relationship between mother and child was profoundly important. In his 1940 work, Bowlby observed that small children should never be subjected to complete or even prolonged separation from their mother [3]. The thought is that the child needs this attachment for reasons of survival. The social environment offered by the mother was seen as the best way of meeting that need [4]. The value of the attachment is that the child learns how the world works, establishing a mental image known as the internal working model [5]. In this, the child has a representation of how the self fits into others with whom the child will relate over a lifetime. This working model is bidirectional in that the child comes to understand how others are perceived, but also how the self is perceived by others [6]. It is through this that the child can extend from the primary relationship with the mother into other relationships, hopefully reinforcing the experiences with the primary relationship.

While developing an understanding of attachment, Bowlby was heavily influenced by psychoanalytic thought, including the works of Anna Freud and Melanie Klein [7]. After WWII, Bowlby joined the Tavistock Clinic in London, which would be his home for many years thereafter. It was here that he connected with James Robertson, a social worker with whom he developed the 1952 film *A Two-Year-Old goes to Hospital*. The film highlighted the intense separation grief when the child did not have access to her mother. This built upon his earlier work The Forty-Four Thieves [2], which consisted of a sample of children referred to a child-guidance clinic. He concluded that “prolonged separation of a child from their mother (or mother-figure) in the early years leads to his becoming a persistent thief and an Affectionless Character (p. 49). His work would become further extended while working with Mary Ainsworth.

In her 1982 paper, Ainsworth [8] described the notion of attachment as a foundation that arose from her reading the work of Konrad Lorenz and Nikolaas Tinbergen. Ainsworth began to develop the notion of the secure base as the successful way in which children develop connections, allowing them to predict safety in relationships [7]. Holmes described Bowlby as concluding that maternal deprivation, particularly if the child is then raised institutionally, will have long-term, serious impacts on the physical, intellectual, social, and emotional development [7] (p. 27), which will lead to an insecure attachment. He believed that the damage that would be caused by removal from maternal care was such that it should be avoided whenever possible.

Mary Ainsworth and Mary Main would extend Bowlby’s work by establishing four patterns of attachment, secure, avoidant, resistant, and disorganized. Attachment styles are linked to the working models of the self and represent the ways in which children exposed to healthy and unhealthy early relationships later come to experience intimate relationships. Children with secure bases tend to be able to shift relationships using both the internal working model and the attachment style [9]. Main et al. [10] raised a series of misconceptions that are unsupported by Attachment Theory. Rosabel-Coto et al. [11] cited three in particular that are relevant to this work:An adult needs to have been present from the infant’s birth in order for an infant to form a secure attachment to that adult;The window of opportunity for the formation of a secure attachment endured only throughout the first three years of life;The amount of time spent with a child is the most important element in forming an enduring attachment relationship (p. 338).

Attachment Theory suggests that the child uses the internal working model of how relationships work to expand into other relationships, expecting that they will work in essentially the same way as the primary relationships. It takes into consideration how others are meant to be available to a child and that the child is worthy of care and attention that is supportive. This then leads to a framework of how the child can anticipate the functioning of relationships external to the family [12,13,14,15,16]. We are not aware of any literature that states that this internal working model must develop solely within dyadic relationships.

As Keller and Chaudhary [17] outlined, Attachment Theory became based upon family life and relationship dynamics that were circumscribed in specific cultures (p. 109) which were those, primarily Eurocentric, where the mother was considered central. Thus, there was a monotropy in attachment theory that built upon the notion of the primary foundational relationship noted above. This was a time when the nuclear family of two married parents in a heterosexual relationship, raising their biological children, was viewed as the ideal structure from which healthy attachment would develop. Attachment Theory frames family from a normative nuclear structure. This lends to an interpretation that the child is fully reliant upon that figure for safety, survival, and the elimination or management of risks in the child’s world [4]. However, in collectivistic or communal environments, such an argument cannot be sustained. Kinship and other community carers play a vital role in raising the child, which runs counter to the Eurocentric notion that it is only the nuclear family that can meet the secure base needs of the child [18,19]. Therefore, the child can develop a secure base through a system of caregivers, rather than through a dyadic mother–child formula. However, much of the literature is not based on cross-cultural attachment patterns that are not based upon the dyadic model [4]. This may partially be the result of the failure of attachment research and Eurocentric professions to develop and apply assessment approaches rooted in a variety of diverse cultural expressions of parenting [17]. Kenkel, Perkeybile, and Carter [20] brought forth the notion of alloparenting, which means that successful care can be provided by those other than biological parents, which can include grandmothers, uncles and aunts, older siblings, fathers, as well as kinship who are communally, rather than biologically, linked to the child. This means that children can experience nurturing care from several people who share the task of raising a child as normative.

This is not to say that children in various cultures do not experience attachment, but that attachment is experienced differently across cultures with a variety of child-rearing models. Keller and Choudhary [17] described five elements of attachment that can be thought of as common across environments:Universality in that children will become attached to one or more caregivers;Secure attachment is normative, but appears differently across cultures;Attachment relies upon sensitive and responsive parenting by caregivers;Child-rearing has no common standard, but varies across cultures; with sensitivity and responsiveness meaning quite different things in distinct cultural contexts;Competence in parenting may look different, and successful parenting outcomes are also different across cultures (pp. 134–135).

There are pluralistic approaches to successful parenting [21]. Attention is also needed on myths that continue to lead to harmful understandings of Attachment Theory, which include an adult needing to be present from an infant’s birth for an infant or child to develop a secure relationship with a new figure [11]. This is quite important as it then leads to a focus on how new people might be introduced to a child to develop a secure relationship as opposed to the notion that it cannot be done. This has implications for foster care. If the transfer of security was not possible, then foster care would be a failure, and so would returning a child to familial or kinship care.

## 3. Indigenous Children in Care

In Canada, Indigenous children are more likely to be in care as opposed to other populations. The 2016 Census indicated that Indigenous children under the age of 14 were 7.7 of the population group, but represented 52.2% of children in care [22]. Many of these children are residing in permanent care outside of their culture. This is related to the colonization of Indigenous peoples in Canada across generations. Placement decisions have historically relied upon providing the best interests as understood from a Eurocentric perspective of family and the raising of a child. To reconsider the question of long-term placements of Indigenous children, consideration must be given to the historical place of Indigenous children. Application within the courts will be considered later in this paper.

To give context, there are more than 630 First Nations communities in Canada, with 50 different identified nations and languages. There are also Metis nations and Innuit peoples in Canada [23]. Thus, there is no pan-Indigenous identity, nor a pan-Indigenous method of parenting.

Indigenous peoples were subjected to a variety of policies that impacted parenting practices within nations and communities. These include the Indian Residential Schools (IRS), which operated from 1876 to 1996 and had mandatory attendance from 1894 to 1947. The sexual, emotional, and physical abuses in IRS were extensive and are now well documented [1]. In addition, an unknown number of children died in these schools. Currently, unmarked graves are being identified at IRS sites, with an estimated 3213 children having died, although this is a conservative estimate. Many children who were quite ill were sent home and died there. Thus, they are not included in the estimate [24]. Hamilton also noted that Dr. Peter Bryce, the chief medical health officer of Canada in the early 1900’s, estimated the death rate related to the IRS to be 8000 per 100,000, while that in the general population overall in that period was 430 per 100,000 (p. 4). The United States has identified similar concerns with the Federal Indian Boarding Schools [25].

Starting in 1951 after amending the *Indian Act* [26], Canada gave responsibility for all Indigenous child protection to provincial governments. Following this, the period of overrepresentation (and over-surveillance) of Indigenous children in care began. The overlap saw Indigenous children apprehended and being put into IRS, followed by a gradual shift to foster care. As this trend continued, the Sixties Scoop (although referred to as the Sixties Scoop, the removal of Indigenous children to child protection care actually began in 1951, although it gained very significant momentum in the 1960s. Children were removed from parental care based upon Eurocentric standards of parenting. However, the removal was strongly connected with Canada’s objective to assimilate Indigenous people into western culture. This practice persisted into the 1980s, with an estimated total of 20,000 children impacted. This has led to the continued overrepresentation of Indigenous children in care [1]) occurred which saw the large-scale removal of Indigenous children, resulting in the placement and adoption of children into non-Indigenous homes. This overrepresentation trend continues without abatement. This historical trend that is still occurring can also be framed as colonially based laws, beliefs, policies, and practices resulting in the over-involvement of the state in the lives of Indigenous people so as to affirm the sustained superiority of colonial society and its beliefs [27,28].

In Figure 1, we show the pattern of colonial involvement since the pre-IRS period until today. As part of this process, the ultimate goal was the full assimilation of Indigenous peoples. When Canada mandated attendance at IRS, the then-Deputy-Minister of Indian Affairs, Duncan Campbell Scott, stated:

“I want to get rid of the Indian problem. I do not think as a matter of fact, that the country ought to continuously protect a class of people who are able to stand alone… Our objective is to continue until there is not a single Indian in Canada that has not been absorbed into the body politic and there is no Indian question, and no Indian Department, that is the whole object of this Bill.” [29]

In response to reports that the Indian Residential Schools were failing to meet the basic needs of Indigenous children [30], Scott noted:

It is readily acknowledged that Indian children lose their natural resistance to illness by habitating so closely in these schools, and that they die at a much higher rate than in their villages. But this alone does not justify a change in the policy of this Department, which is being geared towards the final solution of our Indian Problem. [31]

From this, the foundation of the Indigenous peoples as external to the main direction of Canada becomes evident. As Figure 1 shows, the linkages to Scott’s statements remain.

## 4. Law, Policy, and Assessment

We posit that current policy in social work and other mental health practices, along with assessment tools, can be linked back to the colonization and assimilation policies. This is seen today in definitions within child protection and mental health practices where Eurocentric parenting and family definitions continue to be the most credible [32]. Variations from those principles are considered deficient, if not deviant, from the preferred norm. For example, the literature related to parenting capacity assessments finds its roots in the Eurocentric understanding of family. Tools such as the Parenting Stress Index, which are used as part of assessment, are developed and normed on Eurocentric familial parenting practices. Indeed, the norming of most tools does not provide a vibrant representation of Indigenous populations [33]. This perpetuates the identification of the Indigenous population as not capable, and unable to recover and reclaim culturally based parenting and familial patterns. Offering valid assessment of the best interests of an Indigenous child should start with definitions and practices that are rooted within culture [34]. To do otherwise is to replicate the colonially based power dynamics.

The historical pattern is changeable. Recovery, as seen in Figure 2, occurs within a broad resiliency framework that moves across generations This requires child welfare to think differently about the best interests of the Indigenous child, as noted above. A starting point is for child protection to address the idea that assimilation and removal from culture are in the best interests of the child. It follows then that the child deserves to be a direct part of culture, not living outside of it, nor as a periodic visitor, which is often how placement in non-indigenous homes becomes positioned. This may not occur by ill will, but rather by a continuity of practice that does not, of necessity, determine that “in culture” options are essential for an Indigenous child. To not see the connections to colonization and recovery in Figure 1 and Figure 2 is to deny rights to a child that can be found outlined within the United Nations Declaration on the Rights of Indigenous Peoples (UNDRIP) [35] and the United Nations Declaration of the Rights of the Child (UNCRC) [36]. UNDRIP was adopted as law in Canada, assented to on 21 June 2021 [37].

The essential argument here is that the need to protect a child goes beyond the essentials of life (such as food, clothing, and shelter) to the development of the full human being, which includes the essential elements of who the child is culturally, which is core to their identity.

## 5. Intergenerational Trauma

As has been suggested above, the place of child protection and clinical decisions in the current day are made within the legacy of colonialism and the inter-generational trauma (IGT) that flows from it. There is evidence of trauma transmission across immediate generations and multiple generational trauma within communal and social familial systems [32,34]. These lead to a variety of other negative emotional expressions, as seen in Figure 3. However, caution is required, as IGT can occur within overcoming and thriving, noted in Figure 2, or in the destructive patterns seen in Figure 3. These can bind to the child’s emotional development in safety or in trauma, and in cohesion or in fracturing. Healing also occurs through social transmission. The narrative of child protection leans toward a deficit perspective where, if the IGT has disrupted the child’s progress, then the child needs protection. This often leads to removal from the family, community, and culture, further deepening the divide. Only seeing the impact of IGT denies that strength exists within Indigenous communities, a context that must be applied to raising the child [34].

IGT is treatable [39]. To do so requires an understanding that trauma is a complex issue, as seen in Figure 3. In the case of IGT, we are talking about both complex and chronic traumas, which are often intermixed. For example, the IRS represent both features, as the trauma was multi-faceted (physical, sexual, emotional abuse, malnutrition, and being cut off from family and culture). It was chronic in that it went on over the course of years, as students attended these institutions for several years in a row. Access to families was at the discretion of the school, as seen in Figure 4. This meant that trauma also existed within the family and community systems as a result of the forcible removal of large percentages of children to IRS.

The traumas of the IRS, Sixties Scoop, and the ongoing overrepresentation of Indigenous children in care also link to the left side of Figure 3. Social, policy, and racial pathways combined to transmit the traumas, as the decimation of Indigenous family systems was the goal in order to force assimilation and the desistance of Indigenous peoples [1,40]. The impact of this legacy can be summed up as seen in Figure 5, where social determinants of health have become disrupted due to the IGT, along with the continued legacy of complex and chronic traumas. When a population is systemically traumatized over generations, then disruption in familial, moral, and cultural structures is inevitable.

If the IGT story, as seen in Figure 5, were the end of the narrative, then Canada would have been successful in the assimilation and destruction of Indigenous peoples as unique populations. Canada was not successful, although the overrepresentation noted above serves as evidence of the continuing efforts to separate children from community and family [1].

Further, the emotional response of Indigenous parents to the intrusion of child welfare in their lives triggers long-standing emotions, resulting in poor affect or volatile interaction between the parents and the system. Instead of seeing this as a predictable response, the response itself is used to further support the need for apprehension.

## 6. Place of Attachment Theory

It is common for an Indigenous child to be placed in non-Indigenous care when apprehended from an Indigenous family. This is partly the result of four specific factors:The degree of trauma that is seen as lingering and still active in Indigenous communities is framed as a way to conceptualize the entire community as presenting risk to the child;Due to IGT and IRS, many families in Indigenous communities have a history of involvement in child protection. That history is seen as presenting a risk to the child even when that history might be old or clearly related to assimilation and colonization;It is easier to place a child in a non-Indigenous home, as most established foster placements are non-Indigenous, have been pre-screened, and are available to receive children;As child intervention has been heavily involved in Indigenous communities, those that may care for the child may themselves have prior child intervention involvement. This acts as a barrier for approval as a caregiver.

The practice has often been to place Indigenous children into available foster homes while efforts are made to address the familial parenting issues that brought the child into care. This can take many months or even years. Searches for kinship may be delayed pending the goal of returning the child to their family. If a decision is made to sever parental rights and place the child permanently outside of the family, the child may have been in care for several years. If the parents challenge the decision in court, scheduling delays for trials can significantly lengthen the time the child is in foster care.

Attachment Theory has been used frequently as a basis to determine that a child will be irreparably harmed by removing them from their foster placement and transferring the child to a culturally based kinship placement. The research does not support this conclusion. As noted above, Rosabel-Coto [11] argued that, if a child has developed a secure attachment, then they can manage the change to adoption. The reverse must also be true—that a child can move from a secure placement outside of the family back into the family.

In addition, consideration needs to be given to the harm being done to the child through a failure to afford kinship opportunities. The child’s experience of loss and grief as a result of weakened or fractured connection with culture, community, and kinship systems requires recognition.

All children form attachments, although the quality will vary from child to child and relationship to relationship. These differences will vary in expression, form, and function across cultures. Thus, Eurocentric definitions and assessments are incorrectly applied to Indigenous peoples in Canada [38,39,41,42,43].There is no universally accepted definition of attachment that applies across cultures. Indeed, there is a vibrant research base that shows that attachment variations exist not only in specific cultural contexts, but also that attachment varies from nuclear to communal to collectivistic arrangements [44,45].Attachment Theory was never meant to be used as the basis of child protection decision-making. Indeed, the leading researchers in attachment theory have made that clear [46,47]. Attachment has been used to incorrectly determine that, once a child is placed in a foster (to adopt) or longer-term placement, the child cannot move. Careful consideration has shown that children and adults do create a variety of attachments over the course of their development over a lifetime [48]. Indeed, Duschinsky noted:

Commentators have warned, however, that slippage between the broad and circumscribed use of the term ‘attachment disorder’ has contributed in some quarters to an overdiagnoses of attachment disorders, misuse of appeal to attachment disorder in psychological assessments for family courts and ‘neglect of children’s potential other psychological needs. The gap between clinical discourses and the research paradigm has also been filled at times by inappropriate use of disorganized attachment classification, forced to play the role of a quasi-diagnostic category in child welfare practice. [48] (p. 60)

For social workers, a great deal of teaching in this area has focused upon the dyadic view of parenting in which the child attaches first to a primary caregiver, principally the mother, and then to the next-closest caregiver, typically a father figure. As shown in the Nistawatsiman project [34], this runs contrary to how Indigenous families think of caring for a child. Instead, the Indigenous community thinks of caregiving in a communal sense, where there are many caregivers—aunts, uncles, grandparents, and elders, for example. Thus, a child attaches to many people. This can be seen in other Indigenous cultures, such as the Māori in Aotearoa New Zealand. Atwool [49] stated that “Māori children are not the exclusive possessions of their parents; they belong to *whanau* (extended family), *hapu* (subtribe) and *iwi* (tribe)” (p. 324). Keller added that the expressions of emotion (and thus the emotional bond of attachment) vary “tremendously across cultures” (p. 4). Therefore, how a secure attachment might look will be different across cultures. This does not deny the presence of an important relational network, but illustrates that there is not a single or universal expression of what that may look like [4,50].

In communal societies, connections are, from the beginning, quite different. Connections are multiple, intersectional, and multidirectional. Bowlby, Ainsworth, and Main did not focus upon such views of attachment, although more recent researchers have done so. Keller [4] illustrated the problem by noting that non-Western traditional farmer families socialize infants to follow the directives of caregivers and become part of polyadic social encounters attending to a multiplicity of inputs at the same time. The underlying view of the child is that of a calm, unexpressive, quiet, and harmoniously well-integrated communal agent. Keller et al. [38] concluded, in a comment to the Mesman et al. paper, that Attachment Theory and cultural/cross-cultural psychology are not built on common ground.

Neckoway et al. [43] showed that the dyadic model is not a fit for Canadian Indigenous cultures. This should clearly raise questions as to why the use of the attachment models derived from individualistic Eurocentric cultures continues. As Keller and Bard [45] showed, there are valid ways to think of how collectivistic cultures accomplish attachment and provide for long-term stability and security. There is an error misapplying individualistic interpretations to collectivistic cultural children. It is not to suggest that Indigenous children do not attach—they do. Rather, it is to reconsider that child protection is morally obligated to think of attachment from culturally specific lenses. In Figure 6, this collectivistic notion is illustrated, which acts as a framework for assessing attachment patterns and identifying possible alternative caregivers if a child cannot be with their parents. Elder Roy Bear Chief of the Siksika First Nation describes this as a fundamental traditional belief of his culture [51]. The lesson here is that the Eurocentric understanding of attachment and parenting need not act as the model.

## 7. Place of Attachment Theory

Alloparenting is a concept that is typical in many cultures. Kenkel, Paerkeybile, and Carter [20] defined this as care provided by others, as opposed to parents. Good alloparenting systems may support the greater survival of children. Sear and Mace [53], in a systematic review, indicated that the nuclear family system is less common than might be assumed and suggested, from their research, that human children benefit from extended and kinship support in parenting. Like any form of parenting, this is not to mean that alloparenting is all good and other forms are all bad. Rather, it is to illustrate the importance of not assuming that any one system is “the” system of parenting. Therefore, this also means that Bowlby’s dyadic framework, which Main and Ainsworth expanded upon [5,7,14], should not act as the determinant of what might be in the best interests of an Indigenous child.

Relative to this point is that much of the research relied upon in child protection and mental health using attachment theory is based in a narrow worldview. Critics of the universality of attachment theory have noted that the bulk of research has been on WEIRD—Western, Educated, Industrialized, Rich, and Developed—populations [54,55]. Keller et al. [44] looked at this carefully, as it relates to attachment, showing how this body of research has led to assumptions of universality that are invalid. Bear this in mind as we continue to look at attachment across cultures. They went on to note that “evaluating beliefs and behaviours in one culture according to the standards of another… may be grossly misleading and also unethical” (p. 10). They further stated that “some of the original and core assumptions of attachment theory are not applicable to many cultures around the world” (p. 11). A large body of research now shows that there are many factors that influence how a child develops that are consistent across many, but not all, cultures, whereas there are many factors that are unique to cultures, but there is no single factor that predicts the development of the child [18].

## 8. Psychological Parent

As will be explored later in this paper, the concept of the psychological parent is also drawn upon by the courts. In terms of attachment, it has origins in the work of John and Jean Robertson [56], who were colleagues of Bowlby. They addressed the issue straight on in a way that directly reflects the current controversy:

The phenomenon of attachment and bonding which society welcomes for its binding effect in early adoption is inconvenient in foster care. In recent years we have been made painfully aware that foster parents can become psychological parents and the objects of the foster child’s deepest attachment.

Foster parents have long been expected to keep in mind that their function is only temporary, that they should remain clear about their role and not become ‘possessive’. But no matter how conscientiously restrained a foster mother may try to be, if the child is very young, he will become attached to her and the absent mother will gradually slip into un- importance.[56]

The question of the ‘psychological parent’ can be seen as determinative in case decision-making. The idea finds its origins in the work of Goldstein, Freud, and Solnit in the 1970s [57]. It is worth noting that the work of Goldstein et al. was not based upon clinical research, but the theoretical formulation of the authors. However, even these authors argued that the interests of the child should only be considered to be separate from the family when the family fails to provide nurturing, protection, and affection based upon minimally articulated societal standards. Legal scholars have adopted the concept, particularly in Canada and the United States, but there remains little clinical research to validate the concept.

The psychological parent arises from a Eurocentric understanding [58]. The psychological parent is defined as the building of a close parent-like relationship between a child and another caregiver or primary caregiver. While custody and access issues and divorce matters see the concept applied a great deal, we are most interested in the child intervention and adoption applications. It is being used, for example, in permanent guardianship cases where foster parents apply for adoption, as will be discussed below when looking at case examples in the court.

The original attachment work discussed above felt that there was an overriding benefit of continuity of care to address the child’s sense of time, as well as the physical, emotional, intellectual, social, and moral growth [57] (pp. 31–32), although the work largely lacked multiple family systems views. What was not addressed was the importance of culture and the grounding roots from culture, which might include spirituality and ways of knowing. The Ontario Superior Court of Justice decision *Brown v Canada* [59], arising from the Sixties Scoop, shows that these issues are of significance when deciding on the future of a child. That decision illustrates both the challenges when identity is fractured, as well as the long-term problems faced when attempting to regain a full identity. Thus, an Indigenous view of the meaning of parent and connection is needed.

The attachment and the psychological parent arguments also are relevant in that they pertain very much to the ‘best interests’ tests that courts are often weighing. Most Canadian child welfare legislation makes reference to this test. This matters, as it tends to focus the discussion of the needs of the child as essentially being reliant on the present caregivers. Research also indicates that assumptions that sustaining the present placement are more prone to better outcomes are not supported [60]. The compelling best interest argument is tied to culture in a caregiving environment, in which the child is connected to kin and culture when it is safe to do so.

## 9. Are Children Better off Growing up in Care?

It is difficult to offer a linear, causative answer, as there are many factors impacting the outcomes of children coming into and growing up in care. The research has not generally been hopeful about children who grow up in care. This includes long-term as well as short-term research. For example, Gypen et al. [61] reflected a theme that has been seen in the research fairly consistently: “One of the main findings of this study are the low educational outcomes of foster care alumni. Nearly all researchers found that former foster youth had lower educational attainments than peers from the general population” (p. 81). There is also research that shows that post-secondary educational attainment coming out of care is quite poor [62]. Other research reports higher levels of involvement with criminal justice, mental and physical health systems, unemployment, early pregnancies, homelessness, and other life functioning deficits [63,64,65].

This is not to say that all foster care situations when looked at over the long term are poor. The quality of cohesion within the foster family can make a difference [66]. Other work has supported adoption over foster care [67]. The research overall supports kinship care over other placement choices as yielding generally better outcomes [68]. The data do offer some indications that well-crafted, well supported structural relationships between foster carers and the child’s Indigenous culture can yield good outcomes. Research published in 2020 [69] from British Columbia shows what can be accomplished with highly committed partners, willing to stay in intense connection, working on the basis that connection will be continuous and not episodic. This work also indicates that a stronger connection of an Indigenous foster parent to their own culture also acts as a support for better outcomes. In addition, frequent visits to the child’s nation when the child is not living in proximity is another key element. However, the reader will note the level of commitment needed over time. We are not aware of any significant research that suggests that such a commitment is typical of private guardianship/adoption cases, although such examples do exist. Placement outside of culture works when the risks associated with kinship or cultural placement outweigh those of within-culture placement.

The data do not lead us to conclusions that foster or adoptive care is better. Instead, all things being equal, the data lean toward kinship as being better if biologically immediate caregivers (typically parents) are unavailable.

## 10. Question of Identity

As noted earlier, Indigenous children belong to a nation with a specific connection to history, culture, traditions, and beliefs. The research has helped us see that this is integral to knowing “who” one is within the context of ‘being’ Indigenous [39]. The works referred to earlier regarding the IRS, Sixties Scoop, and Millennial Scoop are all relevant to this issue [1,22,30,34]. A study included a former foster youth who was asked about being raised with her own culture as a Kainai child versus being raised in another Blackfoot nation [70]. She shared her perspective on the various options:

Yes, it would have made a difference given that it is another Blackfoot nation, so I would be learning Blackfoot ways of life and everything, which would be ideal given my circumstances. That would be better than going to a non-Indigenous home, but at the same time that’s not where I grew up. So, ideally, I would want to be with somebody who is either from Kainai, or who lives there, or who lives in a city but is from there and has connections to them. That would be ideal, but secondly, Siksika would be okay given that they are Indigenous. But then, at the same time, I’d still be different because I was from a different reserve…. whether it be somebody that maybe doesn’t have any relation to you, but that’s still like your family, they’re still from where you’re from, and have connections to that nation.[70]

It is perhaps worth knowing that this young lady did have some connection with her family on the nation, but never in a way that sustained her understanding of being a true member of the Kainai First Nation. She described having her feet in two places (white society and Kainai), but never being able to make sense of who she was. She visited her culture as opposed to living it.

In another interview, a Sixties Scoop survivor stated, ”I would have grown up knowing who I was. I had my dad’s side who was all traditional; I could have been exposed right away to culture. I could have known right away who I was. I would have been put in ceremonies. I would have grown up knowing the pipe, the sweat lodge, powwows.” [70].

Umana-Taylor and Hill [71] noted that adoptive parents pay little attention to their children’s ethnic minority backgrounds. Adoptive and foster parents caring for transcultural children typically lack an understanding of what it means to be within another, to live in it, to be of it, and to know it from within the self [32]. Degener et al. [72] found that foster families pay greater attention to the provision of a safe and stable environment and less, perhaps at times no attention, to minority ethnicity. They tended toward silence around the discrimination the children faced, but saw birth parents as connectors to ethnicity. In line with the quotations above, LaBrenz et al. [73] found that implicit bias and structural racism impact placement stability, which may further traumatize and emphasize the losses experienced by ethnic children. Citing Cripps and Laurens, they noted that long-term well-being and resilience came from connections with family, community, and culture. Brown et al. (as cited by [73]) found that, when parents perceived cultural matches with the children placed with them, there were smoother transitions, with the children feeling more secure and less stressed. A strong positive identity, which is linked to the cultural question of who I am, is also linked to better placement stability as well as mental health. This can be achieved through inattentional case planning around racial matching [73] combined with culturally based services, support, and experiences meaningful to the child, their family, and the specific Indigenous community. These experiences need to be persistent such that the child sees connection to their own culture as normative and predictable.

## 11. Case Examples

The longest-standing precedent legal case in Canada regarding attachment and culture is known as *Racine v Woods* [74], which was discussed at length in [70]. The importance to the present paper is that the Supreme Court of Canada, in that case, determined that the attachment relationship that was established with an alternate caregiver was of greater significance than culture. As has been noted throughout this paper, the research on attachment does not bear this out. It is worth noting that the child in the *Racine* case left the non-Indigenous family and ultimately returned to their culture. However, the real challenge for these types of cases is that the trial judge needs to consider the balance of the best interests of the child, bearing in mind the facts laid before the court, and then apply judicial precedents that may help to guide decision-making. This requires that trial judges acknowledge the importance of culture as an equally important component of the definition of best interests.

An example of how attachment theory can be applied in a trial situation is seen in the case of *ZB* [75]. The essential facts are that the child was apprehended at birth and placed into the care of foster parents at the time, i.e., January 2018. The child protection authority was granted permanent care of the child in December of the same year. The maternal extended family was identified in early 2020, although building the relationship with the child was hampered by the COVID-19 pandemic. The foster parents were advised in November 2020 that the child would be transitioning to the kinship placement. In December 2020, the foster parents made an application that would result in the child staying with them, as opposed to transitioning to kinship care. The Court decided to keep the child in the foster parents’ care as private guardians of the child, which had the effect of closing the role of child protection. In making that decision, the Court noted at para 52 “The fact of the matter is that the Director placed the Child with the Foster Parents immediately upon her birth. She has remained there her entire life. She has not been “placed” anywhere else…” The Court added in para 53 “…ultimately require the Court to prioritize the best interests of a child when making decisions relating to that child.” The Court does acknowledge that there is a significant interest in sustaining the child’s connection to culture, but stated “emphasize the importance of Indigenous identity and culture, this is neither the only nor the priority factor to be considered in determining a child’s best interests (Para. 54). The Court determined that placement stability had greater weight than culture given “the best interests of this Child in her particular circumstances.” (Para. 55). This follows very closely the reasoning of the Supreme Court of Canada in the *Racine* decision of 1983 “the significance of cultural background and heritage as opposed to bonding abates over time. The closer the bond that develops with the prospective adoptive parents the less important the racial element becomes” (Para. 187).

We suggest that this line of thinking follows the notion that attachment is primary to a single relationship, not transferable to another party, and follows the dyadic model of Eurocentric nuclear family models, as opposed to the alloparenting of Indigenous communal parenting models.

These cases become substantially more challenging when the child has been in the care of the foster parent for a lengthy period. In the case of *MU* (2022) [76], the Court was faced with a case where a child had been in the non-Indigenous foster home for 12 years and openly expressed a desire to stay there. The child was placed with the foster mother at 5 days of age. The foster mother had actively engaged the child in cultural activities; however, the child’s First Nation sought to differentiate what is essentially visiting culture as opposed to living culture. The court did not accept that line of thinking, stating:

There is also now strong recognition that cultural heritage is an important factor in determining what is in a child’s best interests. Article 20 of the United Nations Convention on the Rights of the Child provides that due regard shall be paid to the desirability of continuity in a child’s upbringing and to the child’s ethnic, religious, cultural and linguistic background. Nevertheless, as our case has clearly shown, there are still contentious questions in law about how much weight to place on a child’s Indigenous heritage when determining what is, in fact, in the best interests of the child.(Para. 222)

However, the Court noted the importance of *Racine* [74], but then stated:

Has the state of the law changed since the decision in *Racine v Woods*? [74] All jurisdictions in Canada have now enacted child welfare legislation requiring judges and agencies to consider a child’s cultural heritage when making a decision regarding the child. The dilemma is how to properly weigh the Indigenous culture as a ‘best interests’ factor.(Para. 227)

The Court noted a series of cases (*SM* [77], *URM* [78], and *DP* [79]) where the best interests of the child relate more to the attachment developed over time in non-Indigenous placement, although the Courts do not dismiss culture, but give it a lower priority. As seen in *MU* [76], this runs counter to the beliefs of Indigenous communities.


**Does culture fade?**


To accept the line of thinking seen in the above case examples, we could expect that attachment does trump culture for the long-term benefit of the child. Thus, cross-cultural placements should be preferable as being consistent with the best interests of the child. In looking at the literature regarding this, there appears to be a strong argument that children seek out and need their cultural place, as it is essential to knowing who they are as a person. In a systematic review, Degener, van Bergen, and Grietens [80] determined that children transracially placed can expect to struggle with their racial/ethnic identity over time. Quite relevant for this paper, they also noted that these children are prone to disconnection with their birth network, although there are exceptions. The authors reported the literature that such placements do impact ethnic identity, although some mitigation is possible with high levels of cultural competency by the foster parents. LaBrenz et al. [73] stated that racial matching matters in terms of placement stability. They also saw the importance of kinship placements.

It is hard to sustain cultural identity when growing up in an environment that seeks to offer placement safety and to operate as a family unit. The foster family, being of a different ethnic origin from the child, tries to follow a balance between the child’s identity and the foster family’s racial identity [72] Foster parents can be receptive to the idea of trying to support cultural identity [81], which matters because the question of sustaining the child’s identity is not one of neglect by the caregivers, but more so of the system that does not see cultural placement as the best and most common option to follow.

In the case of *Brown v Canada* [59], the Court gave careful consideration to the question of whether a child loses their culture or it is a void that the child has an ongoing yearning to fill. This case was focused upon the children who were removed en masse from their Indigenous homes, following which they were placed and adopted in non-Indigenous homes. The breakdown by age 16 of Indigenous children placed in non-Indigenous homes is estimated to be about 85–95% [82]. The Court in *Brown* stated:

The impact on the removed aboriginal children has been described as “horrendous, destructive, devastating and tragic.” The uncontroverted evidence of the plaintiff’s experts is that the loss of their aboriginal identity left the children fundamentally disoriented, with a reduced ability to lead healthy and fulfilling lives. The loss of aboriginal identity resulted in psychiatric disorders, substance abuse, unemployment, violence and numerous suicides. Some researchers argue that the Sixties Scoop was even “more harmful than the residential schools: [59] (Para. 7)

Residential schools incarcerated children for 10 months of the year, but at least the children stayed in an Aboriginal peer group; they always knew their First Nation of origin and who their parents were, and they knew that, eventually, they would be going home. In the foster and adoptive system, Aboriginal children vanished with scarcely a trace, the vast majority of them were placed until they were adults in non-Aboriginal homes where their cultural identity and legal Indian status, their knowledge of their own First Nation, and even their birth names were erased, often forever [83].

The Sixties Scoop is framed as a cultural genocide [1], but it also serves as a stark example of the large-scale impact of non-Indigenous care for Indigenous children. The case also illustrates that these children, around 20,000, did not lose their need for identity rooted within their culture. This acts to affirm that culture does not, in general, fade.

## 12. Discussion

The continued application of a westernized understanding of the way that Attachment Theory is to be applied to Indigenous peoples extends colonial intervention. It is part of a belief that Eurocentric structures, laws, policies, science, and social services all know better than Indigenous peoples about what is needed for their children to be properly cared for. The application of WEIRD science [54,55] also denies that Indigenous peoples have the capacity to assess and respond within their own knowledge systems [32].

Social work has been criticized for grabbing on to Attachment Theory as an evidence-based approach to plan for the best interest outcome for a child. As White, Gibson, Wastell, and Walsh [84] suggested, it has become a method by which to professionalize the profession. It also serves as a way to buffer criticism arising from managerialism oversight when something may go wrong. Theories and evidence-based practices are quite useful when applied appropriately. In this paper, we have argued that social work and the courts may well not be applying Attachment Theory validly with Indigenous peoples. Forslund et al. [46] indicated that Attachment Theory is best used when not attending to individual differences, but rather as a way to work with family to think about the relationships within the family system. We might also think of the theory as being imported into Indigenous populations, rather than it being from their ways of knowing and being, which is something that Joy [85] (p. 188) pondered about in relation to Māori peoples.

Remedies are not easily achieved. Intentional case management that considers not only the cultural needs of Indigenous children, but also the direction of courts becomes central to social work child protection practice. The recent cases of ZB [75] and MU [76] offer a window into change. They have essentially said that, if culture really matters, do not ask the courts to address the issue long after the child has been placed in non-cultural care. The courts are strongly recommending that child protection addresses culture as an immediate priority. Indeed, the decisions reviewed in this paper invite social work and child protection to take the lead in making culture a priority. The alternative is the assumption by the courts that culture is not a central need of a child. This approach should not be seen as diminishing the role of Indigenous-driven case management, but rather as a strong move away from assimilation, which should no longer be seen as a legitimate part of child protection mandate or practice.

We would be remiss if we did not also note that many of the concerns raised in this paper apply to other non-white populations that are also oversurveilled by child protection and where racially inappropriate assessment and intervention approaches are used. Boatswain-Kyte et al. [86] showed that a shift in culture within CPS will be essential to break down racially based biases and be able to build effective working relationships with diverse populations.

## 13. Conclusions

Mental health and child intervention systems intersect with multiple populations whose expression of family varies across cultures. There is no universal definition of family and, thus, there is also no universal ‘right’ way to parent. In this paper, the argument is made that Canadian Indigenous peoples have not only been subject to colonization, but also to systems that try to impose Eurocentric definitions of parenting and child development. This potentially harms children and is a form of adversity placed upon families that can be avoided. Using the examples of how Attachment Theory is misapplied across cultures, professionals and the courts might reconsider how “good enough” parenting might look when seen through the eyes of the culture in which the child is being raised.

## Figures and Tables

**Figure 1 ijerph-19-08754-f001:**
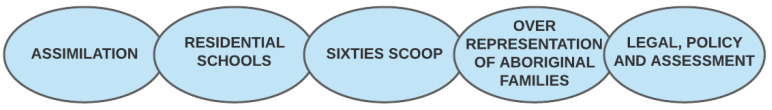
Linkages from historical assimilation patterns to present-day colonial-based child protection.

**Figure 2 ijerph-19-08754-f002:**
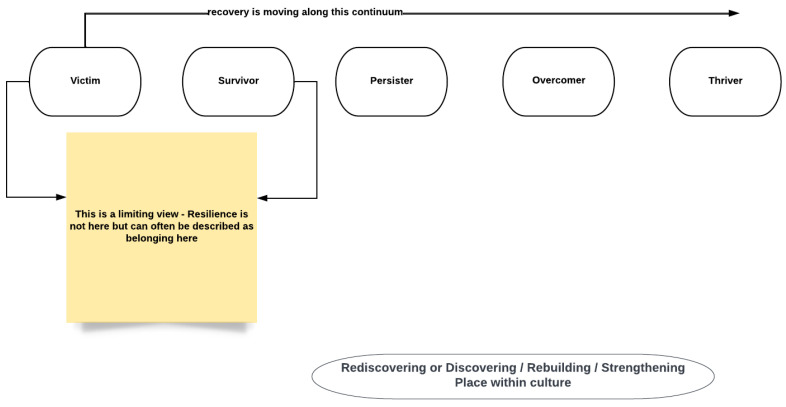
Moving beyond colonial victim/survivor modes both at an individual level, but also at a communal level. Recovery, also framed as success, is seen as progressing beyond the victim/survivor position, moving toward thriving (adapted from [38]).

**Figure 3 ijerph-19-08754-f003:**
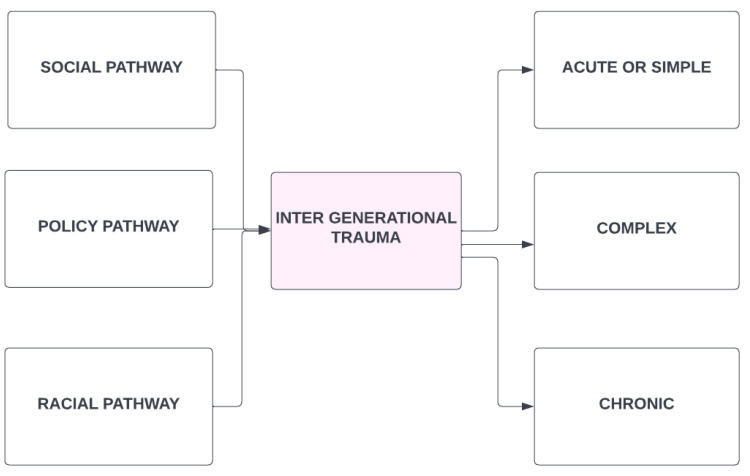
Pathways into inter-generational trauma (IGT) and legacy implications of trauma that may be seen as acute through to chronic.

**Figure 4 ijerph-19-08754-f004:**
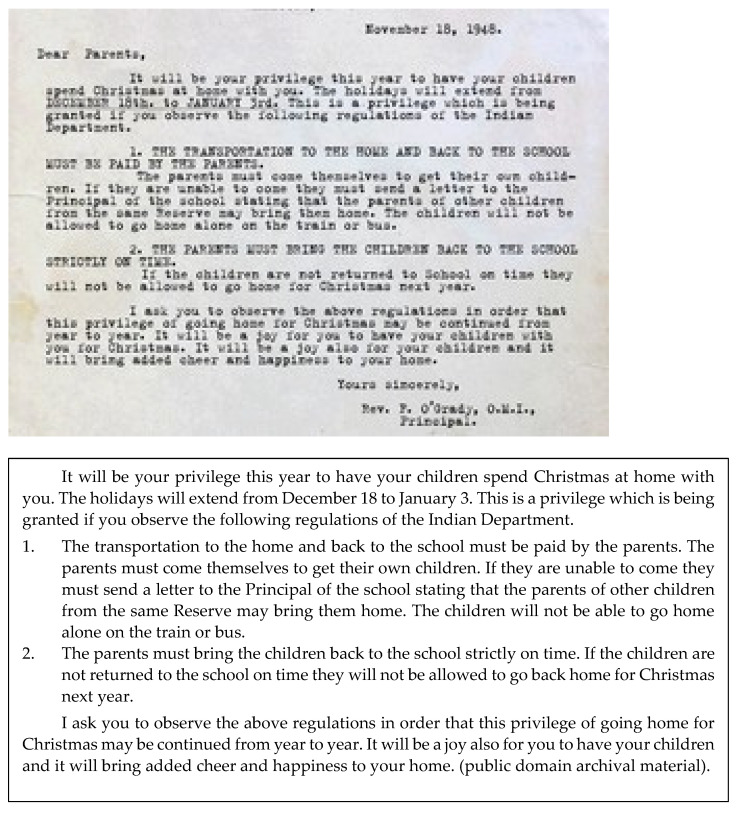
The photograph from the Kamloops Indian Residential Schools dated 18 November 1948 addressed to Dear Parents.

**Figure 5 ijerph-19-08754-f005:**
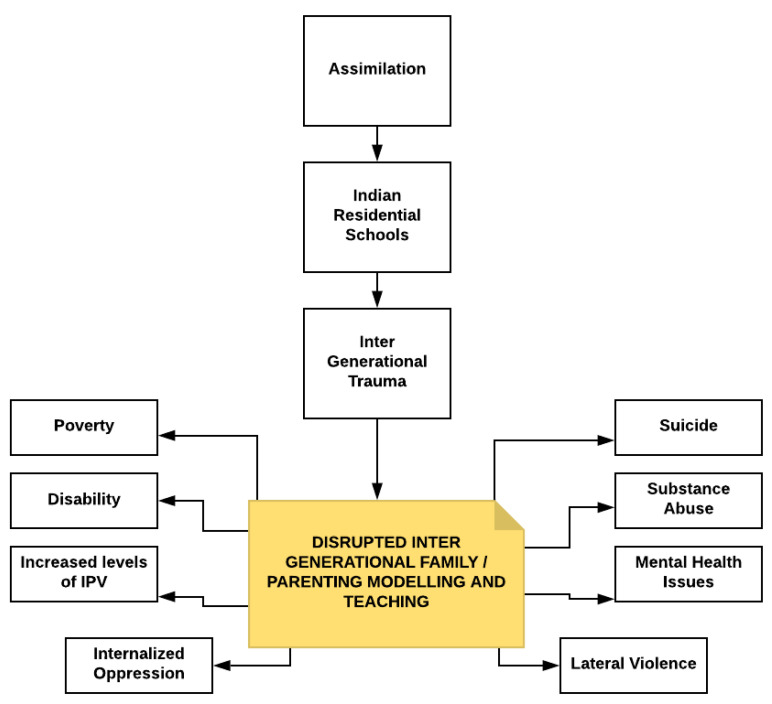
Disrupted patterns in social and familial transmission arising from inter-generational trauma.

**Figure 6 ijerph-19-08754-f006:**
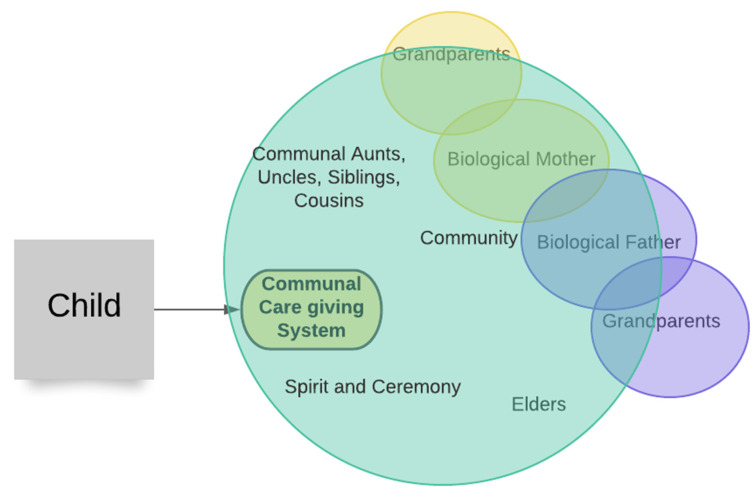
Intersectional view of childcaring within an Indigenous context, which, from early life, creates multiple attachment pathways. Amir and McAuliffe [52] have shown the value of the “in culture” experience as opposed to across-culture experiences. They note ”A general feature of these models is that they all seek in some way to contextualize child development as a dialogue between the individual and the various social, ecological, and cultural inputs they experience” (p. 433). The child is not separate from the cultural context, but an integrated part of it.

## Data Availability

Not applicable.

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
