# Peer review of "Attachment Theory: A Barrier for Indigenous Children Involved with Child Protection"

_ijerph, 2022, doi:10.3390/ijerph19148754_

Round 1

Reviewer 1 Report

Dear Authors,

As a previous CW/adoption worker this article so affirms my experience and my concerns over the overuse of attachment theory to keep children of color from being reunited with birthfamily and or to be placed in homes that honor and help to develop/maintain the child's racial/cultural identity. I look forward to seeing this in print

Author Response

Thank you for the review. Attached is an overview of the changes we have made incorporating  the feedback from all three reviewers.

Reviewer 2 Report

Thank you for the invitation to review this draft. Based on Attachment Theory, this article discusses how the application of attachment theory violates the best interests of Indigenous children with the Canadian examples, which has certain practical significance. However, I believe the paper can be strengthened furthermore if the study addresses some issues as follows:

Abstract:

1. The organization of the abstract is not clear enough. It is recommended that the abstract be described in the format of background, method, result and conclusion.

2. It is suggested to change the structure of the article into "introduction, method, result, discussion and conclusion" to make the article more standard and logical.

Introduction:

The introduction needs to complement the main purpose and significance of the study, and the research status of this research topic.

Attachment Theory:

(Page2,Line83-84) Chilidren with secure bases tend to be able to shift relationships using both the internal 83 working model and the attachment style.

Please pay attention to spelling. "Chilidren" should be changed to "Children".

Law, Policy and Assessment:

The serial number of this subheading is the same as that of the previous subheading, please correct it.

Intergenerational trauma:

Photo 1 is not clear enough, please replace it.

The place of Attachment Theory:

There are two identical subheadings in the article, namely "The place of Attachment Theory". Please change or delete one of the subheadings.

Discussion:

The discussion is insufficient and needs to be improved. The author should summarize the core content of the article and conduct a more in-depth analysis.

Author Response

(The authors gave the same response as above.)

Reviewer 3 Report

The paper presents an argument based partly on previous research evidence and partly on two case examples form law which are used to problematise a particular application of Attachment Theory in child protection cases for Indigenous children in Canada.

The argument is quite strongly put forward, premised on the evidence of a failed child protection system and the apparent outcomes it has produced. While I personally am inclined to agree with the premise and even the conclusions presented, there are parts of the argument that require a leap of faith, and which don't have sufficient evidence to support them. For example, the argument assumes that the adverse outcomes experienced by people who have been apprehended as children are the direct consequence of the child protection system, even though the traumas experienced by people are acknowledged to be intergenerational.

I'm sure the authors would agree that the causal factors leading to adverse outcomes are multiple and complex. So I think the authors should put some caveats on the conclusions drawn. For example, in cases where appropriate kinship care is not available, some form of harm is almost inevitable, but even in cases where kinship care is deemed appropriate, harm is still a possible (if not likely) outcome.

In terms of the application of the Theory, I think the authors need to be more specific. It seems to me they are arguing against a particular reading of the Theory and its legal application. But in their discussion, from section 7 on, they seem to suggest that all applications of the Theory are problematic because Indigenous communities have alternative ways of coming up with solutions to problems of attachment. Now I could be wrong in my reading of the arguments, but it seems to me that the leaps of faith required in accepting the arguments coupled with the generalisations made for all Indigenous communities, children and parenting structures, are open to considerable challenge.

I have made a number of comments and suggestions in markup on the attached file for consideration. I don’t think those comments affect the thrust of the paper, but might help the authors avoid unnecessary criticism from critical readers.

The paper is a solid contribution to the literature and I commend the authors for their work.

Author Response

(The authors gave the same response as above.)
